# Photobiomodulation Therapy in Hypertension Management—Evidence from a Systematic Review and Meta-Analysis

**DOI:** 10.3390/jcm14196716

**Published:** 2025-09-23

**Authors:** Lara Maria Bataglia Espósito, Francisco Costa da Rocha, Praveen R. Arany, Cleber Ferraresi

**Affiliations:** 1Physical Therapy Department, Federal University of São Carlos (UFSCar), São Carlos, 13565-905, SP, Brazil; franciscorocha@estudante.ufscar.br (F.C.d.R.); cleber.ferraresi@gmail.com (C.F.); 2Departments of Oral Biology, Surgery and Biomedical Engineering, University at Buffalo, Buffalo, NY 14214, USA; prarany@buffalo.edu

**Keywords:** low-level light therapy, photobiomodulation, hypertension, high blood pressure

## Abstract

**Background:** Hypertensive patients have difficulties in controlling blood pressure (BP), resulting in high mortality rates. There is a growing number of lab and human studies investigating the effects of photobiomodulation (PBM) therapy on hypertension. This motivated the current work to systematically assess existing literature and group evidence on the utility of PBM in BP management. **Methods:** A systematic review with meta-analysis was performed on randomized clinical trials (RCTs) and experimental animal studies assessing PBM treatments in hypertensive patients/animals. Five primary databases were assessed by two reviewers. Descriptive and quantitative analyses were performed separately for clinical and experimental studies. **Results:** A total of 197 articles were screened that identified four RCTs and five experimental studies for final inclusion. The clinical trials noted that PBM treatments reduced systolic blood pressure (SBP), diastolic blood pressure (DBP), and heart rate (HR), but with very low certainty. Experimental lab studies corroborated that PBM treatments reduced SBP, DBP, and mean arterial pressure (MAP) while increasing nitric oxide levels, again with very low certainty. Overall, RCTs had a high risk of bias, and experimental studies had an unclear risk of bias. **Conclusions:** PBM treatments have the potential to be an adjunct therapy for the treatment of hypertension, with studies showing a possible reduction in SBP, DBP, MAP, and HR, but the evidence is of very low certainty, coming from RCTs with a high risk of bias and experimental studies with an unclear risk of bias. The current evidence needs to be significantly improved with rigorous, well-designed experimental and clinical studies.

## 1. Introduction

Hypertension is a significant chronic health condition characterized by persistent high blood pressure (BP) levels. This disease is the leading cause of death worldwide and is known to be a significant risk factor for cardiovascular diseases, chronic kidney disease, stroke, and premature death [1,2,3,4]. Around 1.28 billion adults are affected by hypertension, and about 10.4 million deaths per year can be associated with high BP—especially in low- and middle-income countries [1,4,5]. Furthermore, besides the direct impact of hypertension on an individual’s health, it also results in a major burden on the economy and society [2]. Current well-established treatments for hypertension that use pharmacotherapy, diet, and lifestyle changes remain poorly effective [5,6]. Globally, the effectiveness of routine management of hypertension is as low as 23% for women and 18% for men [6]. Low adhesion to established therapies is one of the big reasons for the poor control of BP [3,4]. Therefore, novel therapies improving BP control could benefit the vast patient population struggling with this chronic disease.

Among various approaches, the use of low-dose light treatments, termed photobiomodulation (PBM) therapy, has been explored [7,8,9,10]. PBM aims to modulate biological processes such as pain, inflammation, and tissue repair [11,12]. There has been significant recent progress in our understanding of PBM mechanisms. Light is absorbed by chromophores, mainly the cytochrome c oxidase. Absorption of light triggers a series of beneficial cellular responses, such as increased mitochondrial membrane potential, adenosine triphosphate (ATP), and reactive oxygen species (ROS) and the photoactivation of an extracellular latent growth factor, TGF-β1 [11,13,14]. Recent studies on the therapeutic benefit of PBM on hypertension have garnered some attention. A primary hypothesis motivating these studies is the ability of PBM treatments to increase nitric oxide (NO) levels in blood vessels, since promoting vasodilation could reduce blood pressure [7]. Current clinical studies have focused primarily on the effects of PBM on systolic (SBP) and diastolic blood pressure (DBP) [15].

Progress in these mechanistic aspects and improvements in photonics technologies have enabled more stringent attention to device and treatment parameters, such as wavelength (nm), photon energy (eV), power (W), power density (W/cm^2^), irradiation time (sec), energy (J), energy density (J/cm^2^), photon fluence (p.J/cm^2^), and Einstein (E) [16]. An important aspect of PBM treatment is the site of treatment. There has been much debate on global (whole-body) versus specific (focal) treatments. The field of laser acupuncture has developed in parallel, where PBM is performed on specific acupuncture points. This has been a particularly popular approach with PBM investigations in randomized controlled trials (RCTs) to manage hypertension [15,17]. While several clinical and lab research studies exist, there is a lack of a systematic review and meta-analysis of literature, which motivated the current work. Meanwhile, the search for new therapeutic strategies for hypertension is essential. There has been a growing number of articles in this area, indicating a recent interest in the topic. Furthermore, PBM parameters and application points lack standardization. This highlights the need for a systematic synthesis of this topic, compiling the current evidence for clinical and animal studies.

The purpose of this work was to systematically evaluate the available evidence about the effects of PBM therapy on the BP of hypertensive individuals and hypertension animal models and to identify gaps in knowledge for future clinical and/or experimental studies.

## 2. Materials and Methods

This systematic review was conducted in line with the Cochrane Handbook for Systematic Reviews [18] and described according to PRISMA (Preferred Reporting Items for Systematic Reviews and Meta-Analyses) [19] and SYRCLE (Systematic Review Centre for Laboratory Animal Experimentation) [20] recommendations. The protocol for the systematic review was registered in the International Prospective Register of Systematic Reviews (PROSPERO) in May 2024 (registration number: CRD42024530019).

### 2.1. Eligibility Criteria

Only randomized controlled trials (RCTs) or experimental studies with rats or mice that directly assessed the effects of PBM treatment (laser or light-emitting diode, LED) effects on blood pressure in diagnosed hypertensive individuals or hypertensive animal models that were compared to a placebo or control group were included in this analysis. RCTs were the design of choice for clinical studies due to their lower likelihood of being biased and influenced by confounding factors [18].

Studies with individuals of both sexes, with no limits of age, that assessed acute (one session) or chronic effects (more than one session) of PBM treatments were included. In vitro (cell culture) studies, observational studies, case studies, systematic reviews, integrative reviews, abstracts from congresses, dissertations, theses, unpublished studies, and studies with no comparison group were all excluded. In addition, articles without full-text and studies that could not be translated were also excluded. In addition, studies that combined PBM treatments with other therapies other than the routine drugs for hypertension were excluded. Studies that did not assess BP were not included.

### 2.2. Outcomes

The outcomes of interest were systolic blood pressure (SBP), diastolic blood pressure (DBP), mean arterial pressure (MAP), heart rate (HR), nitric oxide (nitrite and nitrate) values, and duration of the observed effects. We also considered potential adverse effects related to PBM treatments reported by the authors.

### 2.3. Search Strategy

A systematic search without language or date restrictions was conducted on 5 databases: PubMed, Embase, Cochrane Library (Central Register of Controlled Trials), CINAHL, and Web of Science. Manual searches in the references list of included articles were also conducted. The initial search was carried out on 10 April 2024 and was updated on 14 August 2025. The search strategy used for PubMed and adapted for other databases combined the most common terms and its synonyms used to identify the population (hypertensive individuals or rats or mice submitted to a hypertension model), the intervention (PBM), and the type of study (RCT or experimental study) with Boolean operators. The search strategy can be found in Appendix A.

### 2.4. Selection

Two independent reviewers carried out the study selection, starting with title analysis, followed by abstract, and full-text assessment, as per the eligibility criteria in all phases. Disagreements were resolved by discussion between reviewers analyzing each of the established criteria. If there was no consensus, a third reviewer was responsible for the decision. References were downloaded from the databases and uploaded to the Rayyan website (http://www.rayyan.ai/, accessed on 10 April 2024) to conduct the selection.

### 2.5. Data Extraction

A standardized collection form was used to collect data from included studies. The following data were collected: type of study (RCT or experimental); study design; participant or animal characteristics; type and grade of hypertension or hypertension model; PBM treatment characteristics and dosage (source of light, equipment, wavelength, energy, energy density, power, power density, spot size, number and location of points, irradiation time, irradiation technique, and number of sessions); assessed outcomes; data collection; outcomes analysis; mean, standard deviation, and number of participants in each group for the assessed outcomes; and adverse effects. If there were any missing data about the study, especially data for the construction of meta-analyses, authors were contacted via email or by any means of contact that were available online. If the data were not made available, the study was not included in meta-analyses.

### 2.6. Assessment of Risk of Bias

Two independent reviewers carried out a risk of bias assessment for all included studies. For RCTs, version 2 of the Cochrane tool for assessing risk of bias in randomized trials (RoB 2) [21] and the PEDro (Physiotherapy Evidence Database) scale [22] were used. The RoB 2 tool considers five bias domains: randomization process; deviations from intended interventions; missing outcome data; measurement of the outcome; and selection of the reported result. Each domain can be classified as low risk of bias, high risk of bias, or some concerns. The domains were classified for each study by answering signaling questions provided by the tool and then judging risk of bias based on the answers [21]. An overall low risk of bias means that all domains were judged to be at low risk of bias. An overall judgment of some concerns means that at least one domain was judged to raise some concerns, but not at high risk of bias, and an overall high risk of bias means that at least one domain was judged to be at high risk of bias.

The PEDro scale evaluates the methodological quality of RCTs based on internal validity, external validity, and statistical analysis considering eleven criteria. Each one of the met criteria is punctuated with “1” or “0” if the criteria were not met. The final score can be obtained by adding criteria from 2 to 11, totaling from 0 to 10 points. The higher the punctuation, the better the methodological quality. An excel sheet was constructed to punctuate the criteria for each included study, guided by explanations of each criteria available in the PEDro scale website [23].

For experimental studies, the SYRCLE’s risk of bias tool (SYRCLE’s RoB tool) was used. The SYRCLE’s RoB tool is based on the Cochrane Collaboration RoB tool, but it was adapted to experimental animal studies, considering aspects of bias that are specific to this type of study. The tool comprehends different types of bias: selection bias; performance bias; detection bias; attrition bias; reporting bias; and other identified sources of bias. The items were classified as low, unclear, or high risk using signaling questions provided by the tool, guided by the description of the domains by SYRCLE’s authors [24].

For all scales, disagreements were resolved by discussion between reviewers analyzing each of the established criteria. If there was no consensus, a third reviewer was responsible for the decision.

### 2.7. Data Analysis and Synthesis

Clinical studies and experimental studies were analyzed separately. Descriptive and quantitative analyses were conducted from the data extracted from included studies. Analyses were also performed separately for articles that studied the acute and chronic effects of PBM treatments. The main data extracted from the included studies were presented in tables. Meta-analyses were conducted for all the outcomes that presented enough data from the studies, separating study type (clinical and experimental) and the different time effects of PBM treatments (acute vs. chronic). Mean and standard deviation values were used for continuous data, and since the studies used the same units of measure, the results were presented as the mean difference. The confidence interval (95%) and heterogeneity, using the I^2^ statistic, were calculated for all meta-analyses. The random effects model was selected assuming variation across studies. Funnel plots and subgroup analyses were not possible to perform due to the small number of studies included in each meta-analysis. Sensitivity analyses considering PBM wavelengths were performed for the outcome SBP in experimental studies, as this was the only meta-analysis that included a sufficient number of studies with distinct wavelengths.

Review Manager 5.4. software (Cochrane IMS, Copenhagen, Denmark) was used for all the quantitative analyses.

### 2.8. Summary of Findings and Assessment of the Certainty of the Evidence

The certainty of the evidence was assessed by the GRADE (Grading of Recommendations, Assessment, Development, and Evaluations) system. Specific recommendations were followed for grading the preclinical animal intervention studies [25]. The GRADE system is used for rating the quality of a body of evidence in one of four categories: high, moderate, low, or very low. The items to be considered for grading the evidence are: risk of bias, related to flaws in the design of individual studies; imprecision, related to the risk of random error within the body of evidence; inconsistency, related to systematic differences across study results in an evidence synthesis; indirectness, related to differences about the comparators between studies that can affect the applicability of the body of evidence; and publication bias, related to the possibility of only studies with positive results being published [26,27]. The GRADE system was used to estimate the effect for each one of the assessed outcomes.

## 3. Results

The database search identified a total of 197 articles. Considering the eligibility criteria, four randomized controlled trials (RCTs) and five experimental studies were finally included in this review (Figure 1). Appendix A contains articles excluded in the abstract, full-text analysis, and the reasons for exclusions.

### 3.1. Characteristics of Randomized Controlled Trials

The RCT sample sizes ranged from 45 to 79 participants, totaling 229 hypertensive participants. Two RCTs were conducted in Egypt, one in Brazil, and one in the United States. Patients’ ages ranged from 20 to 75 years old. Two studies included both men and women [15,28], one included just men [29], and one included just women [17]. The range of blood pressure varied between studies. Three studies included patients with a blood pressure higher than 140/90 mmHg. Others included levels up to 160/100 mmHg [29] and 170/105 mmHg [17], while another did not establish an upper limit level [15]. The other study considered blood pressure levels from 125/81 to 160/110 [28]. Four studies assessed chronic effects of PBM, one with 4 weeks of therapy [29], two with 6 weeks [15,17], and one with 12 weeks [28]. The characteristics of the included RCTs are presented in Table A1.

### 3.2. Characteristics of Experimental Studies

Considering experimental studies, the sample sizes ranged from 16 to 42 rats, totaling 134 animals. All five studies were conducted in Brazil, and only one included female rats. Three studies used a spontaneously hypertensive rat (SHR) model [7,8,10], while one study used obese rats with hypertension induced by a high-fat diet [30], and the other one used an ovariectomized model [31]. Only one study established a blood pressure level to consider the animals hypertensive, including animals presenting an SBP higher or equal to 160 mmHg [7]. Four studies investigated the chronic effects of PBM treatments, one with 2 weeks of intervention [31], another with 4 weeks [7], another with 7 weeks [10], and the last one with 12 weeks [30]. One study assessed the acute effect of PBM treatment [8]. The characteristics of the included experimental studies are presented in Table A2.

### 3.3. Outcomes—Randomized Controlled Trials

#### 3.3.1. Systolic Blood Pressure

All included studies measured systolic blood pressure in millimeters of mercury (mmHg). One study assessed BP by a sphygmomanometer [17], another by a digital upper-arm BP monitor [29], another described that BP was measured using the indirect oscillometer method [15], and the last one measured BP by a manual BP monitor [28]. One study compared laser acupuncture plus Captopril-25 mg (ACE inhibitor) to Captopril-25 mg twice daily only (without laser) for 6 weeks and found that laser acupuncture significantly decreased SBP, compared to the antihypertensive alone [17]. Another study compared 4 weeks of laser acupuncture to a control group (no intervention or drugs) and found reduced SBP correlated with laser acupuncture treatments [29]. Two studies compared laser acupuncture to sham therapy. One of these studies carried out 6 weeks of therapy in patients who used drugs for hypertension for at least one year [15]. The other study performed 12 weeks of therapy in patients who did not use a drug for hypertension [28]. Both studies found reduced SBP in the laser acupuncture group. However, post-intervention comparisons between groups were not performed in these two studies. The meta-analysis result, including three studies (*n* = 169), showed that when compared to control, sham therapy, or antihypertensive drugs, the PBM treatments had a reduction in SBP with a mean difference of −15.87 mmHg (confidence interval = −27.11 to −4.64 mmHg; *p* = 0.006; Figure 2), with very low-certainty evidence and considerable heterogeneity (I^2^ = 92%).

#### 3.3.2. Diastolic Blood Pressure

All included studies measured diastolic blood pressure in millimeters of mercury (mmHg). One study compared 6 weeks of laser acupuncture plus Captopril-25 mg to Captopril-25 mg twice daily only and found decreased levels of DBP related to laser acupuncture when compared to the antihypertensive treatments alone [17]. Another study compared laser acupuncture to a control group (no intervention or drugs) and also found reduced DBP related to laser acupuncture after 4 weeks of treatment [29]. Two other studies compared laser acupuncture to sham therapy, one in patients using drugs for hypertension control for at least one year [15] and the other in patients who did not use a drug for this disease [28]. Both studies found significantly reduced DBP levels related to laser acupuncture, one after 6 weeks of therapy [15] and the other after 12 weeks [28]. However, these two studies did not compare the post-intervention values between groups. The meta-analysis result, including three studies (*n* = 169), showed that when compared to control, sham therapy, or antihypertensive drugs, PBM treatments had a reduction in DBP with a mean difference of −8.73 mmHg (confidence interval = −15.19 to −2.26 mmHg; *p* = 0.008; Figure 3), with very low-certainty evidence and considerable heterogeneity (I^2^ = 94%).

#### 3.3.3. Heart Rate

Heart rate was assessed in only one of the included studies [29]. The authors assessed HR by a digital upper-arm BP monitor in beats per minute (bpm). Laser acupuncture was compared to a control group (no intervention or drugs) for 4 weeks of treatment and was associated with decreased levels of HR, compared to the control group, with very low-certainty evidence. In the post-treatment analysis, the laser acupuncture group had a mean HR of 76.9 ± 4.50 bpm, while the control group had a mean HR of 80.2 ± 8.30 bpm [29].

#### 3.3.4. Adverse Effects

None of the included studies reported any adverse effects related to PBM treatments, but no study included specific investigation methods for adverse effects.

### 3.4. Outcomes—Experimental Studies

#### 3.4.1. Systolic Blood Pressure

All included studies measured systolic blood pressure in millimeters of mercury (mmHg). Two studies measured the hemodynamic parameters by catheter implantation in an artery [8,10] and three by the tail-cuff plethysmography technique [7,30,31]. One study compared PBM treatments to sham therapy with 7 weeks of treatment and did not see a reduction in SBP related to PBM [10]. Two other studies compared PBM to sham therapy, one with 4 weeks of irradiation [7] and one with only one application (acute effect) [8]. Both studies identified a reduction in SBP, and the acute study showed that the duration of the hypotensive effect was around 25 min [8]. However, the reduction in these two studies was only seen in the so-called (by the authors of the article) “PBM-responsive animals”. One study compared PBM treatments with high-fat diet to a high-fat diet-only group (control group). PBM treatments prevented increases in SBP induced by the high-fat diet [30]. Another study compared ovariectomized rats with ovariectomized rats treated with a 2-week PBM protocol. The treatment with PBM induced a statistically significant decrease in SBP when compared to the ovariectomized rats that did not receive PBM [31].

The meta-analysis result, including three studies (*n* = 49), showed that when compared to control or sham therapy, PBM treatments had a reduction in SBP with a mean difference of −14.44 mmHg (confidence interval = −18.10 to −10.79 mmHg; *p* < 0.00001; Figure 4), with very low-certainty evidence and moderate heterogeneity (I^2^ = 43%). Sensitivity analysis of this outcome based on the wavelength showed that the exclusion of Tomimura et al. [10], the only study that used an infrared wavelength, reduced heterogeneity to 0%, while the direction of the effect size remained the same.

#### 3.4.2. Diastolic Blood Pressure

Two included studies measured diastolic blood pressure in millimeters of mercury (mmHg). One study (7 weeks of therapy) compared PBM to sham therapy and showed that the laser group had a reduction in DBP. The laser group resulted in a DBP of 143 ± 4 mmHg, while the sham group DBP was 157 ± 3 mmHg [10]. The other study also compared PBM to sham therapy but with only one application. The authors showed that PBM treatments reduced DBP by a magnitude higher than the sham in responsive animals (58% of the rats—criteria defined by the authors). The mean differences considering pre- and post-intervention values for PBM and sham groups were −9.6 ± 2.7 mmHg and −1.5 ± 2.2 mmHg, respectively, only considering responsive animals, with the effect lasting for about 25 min [8]. The certainty of the evidence was graded as very low.

#### 3.4.3. Mean Arterial Pressure

Two included studies measured mean arterial pressure (MAP) in millimeters of mercury (mmHg). One study compared PBM treatments to sham therapy with 7 weeks of application and showed that the therapy reduced MAP. The PBM and sham groups had MAPs of 169 ± 4 mmHg and 182 ± 4 mmHg, respectively [10]. The other study, which assessed the acute effects of PBM to sham therapy, also observed positive effects, noting reduced MAP of a higher magnitude than the sham treatments in responsive animals (65% of the rats—criteria defined by the authors). The mean differences considering pre- and post-intervention values for PBM and sham groups were −11.0 ± 3.73 mmHg and −1.4 ± 1.87 mmHg, respectively, only considering responsive animals. The hypotensive effect lasted for about 25 min [8]. The certainty of the evidence was graded as very low.

#### 3.4.4. Heart Rate

Two studies assessed heart rate in beats per minute (bpm). One study compared PBM to sham therapy and showed that HR at rest was reduced in the treated group after 7 weeks of intervention (312 ± 14 bpm and 361 ± 13 bpm for the PBM and sham groups, respectively) [10]. The other study also compared single PBM to sham treatments and did not observe any differences in HR (mean differences of −8.69 ± 6.52 bpm and −6.74 ± 4.63 bpm for the PBM and sham groups, respectively) [8]. The certainty of the evidence was graded as very low.

#### 3.4.5. Nitric Oxide (Nitrite and Nitrate)

Four studies assessed nitric oxide production by blood analysis. Three of these studies assessed both serum nitrite and nitrate (NOx) [7,30,31], while one assessed only nitrate [8]. One study examined PBM + high-fat diet compared to a high-fat diet-only group (control group) for 12 weeks and found that PBM treatments normalized NO concentration in obese animals [30]. Another study compared a 2-week PBM treatment in ovariectomized rats with ovariectomized rats that did not receive this treatment on NOx production. The authors found that PBM treatment resulted in an increased concentration of NO [31]. Other authors compared PBM to sham therapy for 4 weeks but did not see a difference in NO 24 h after the last treatment. These values could not be added to the meta-analysis because the authors reported the results from responsive and non-responsive animals separately (criteria defined by the authors) [7]. Another study compared a single PBM treatment to sham therapy on nitrate production. The authors identified an increase in nitrate related to PBM therapy in responsive animals that also presented reduced BP levels [8].

The meta-analysis results, including two studies (*n* = 33), showed that when compared to control or sham therapy, PBM treatments induced an increase in NOx with a mean difference of 12.63 μM (confidence interval = 2.97 to 22.28 μM; *p* = 0.01; Figure 5), with very low-certainty evidence and considerable heterogeneity (I^2^ = 94%).

#### 3.4.6. Adverse Effects

None of the included studies reported any adverse effects related to PBM treatments, but no study included specific investigation methods for adverse effects.

### 3.5. Photobiomodulation Therapy Protocols

#### 3.5.1. Randomized Controlled Trials

All four included RCTs carried out laser acupuncture protocols. A summary of the PBM protocol parameters in RCTs is described in Table A3. Two of them performed sustained protocols of 6 weeks, one with three sessions per week [17] and one with one session per week [15]. Another study carried out a sustained protocol of 4 weeks, with sessions three times per week [29]. The other study performed two sessions per week for 12 weeks [28]. Three of the studies used infrared wavelengths, and one described it as 904 nm [29], while the other two did not specify the wavelength [15,28]. One study did not report the wavelength [17]. Among the four studies, only one study reported that the total energy (J) used in the treatment was 16 J (8 J per point) [28]. The other three studies did not provide sufficient parameters to assess the dose. As for power (mW or W), one study reported that the average peak power used was 5 mW, with a frequency of 5000 Hz [29]. One study only reported the power as 6 mW [15]. Another study reported a frequency of 10 kHz, with an average power of 33 mW for 8 min (480 s) for a total energy of 16 J [28]. The other studies did not report enough parameters to estimate the power (mW or W) [17].

Two of the studies applied the therapy to two specific acupuncture points, one with 120 s of treatment time [29] and the other with 240 s for each point [28]. Another study applied the therapy to 3 acupuncture points with an irradiation time of 240 s per point [17], and the last one used 11 acupuncture points, with a total irradiation time of 1440 s [15]. All studies applied laser acupuncture to the LI 4 and LI 11 points, located in the thumb and elbow regions. One study also applied the therapy to the Spleen 6 point, located in the leg [17], and Pereira et al. [15] also applied points located in the head and lower limbs.

#### 3.5.2. Experimental Studies

Four of the included experimental studies carried out sustained protocols of PBM treatments. Details of the PBM protocols used in these experimental studies are summarized in Table A3. The first one carried out a 2-week protocol of PBM with irradiations twice a week [31]. Another one performed 4 weeks of PBM treatments, with two sessions per week [7]. Another study carried out 7 weeks of PBM treatments three times per week [10], and others performed the therapy for 12 weeks, also three times per week [30]. The other study investigated the acute effect of PBM treatments with one session [8]. Four of the included studies used a red wavelength of 660 nm [7,8,30,31], and one study used an infrared wavelength of 780 nm [10]. Four studies used an energy of 5.6 J per point [7,8,30,31], and one used an energy of 3.6 J per point (calculated by the authors of the review) [10]. Four studies used a power of 100 mW [7,8,30,31], and one used a power of 40 mW [10]. Two studies treated the animals’ tail transcutaneously, one at three different points with 56 s per point [7] and the other with only one point for 90 s [10]. The three other studies treated the animals’ abdominal region at six points (in contact). One of them treated the points separately [8] for 56 s per point, while other treated six points simultaneously for 56 s [30], and the other one did not report it [31].

### 3.6. Risk of Bias

Risk of bias for RCTs assessed by RoB 2 resulted in an overall high risk of bias for all included studies (Figure 6). The most concerning domains were bias due to deviations from intended intervention and bias due to missing outcome data. Methodological quality scores for RCTs assessed by the PEDro scale ranged from 4 to 7 in a scale from 0 to 10. One study had four points [29], two had five points [17,28], and one had seven points [15]. The main reasons for losing points were lack of concealed allocation, blinding of therapists, blinding of assessors, sample loss, and intention to treat analysis. The PEDro scale assessment can be found in the Appendix A.

Methodological quality for experimental studies assessed by SYRCLE’s risk of bias tool had classifications of low, high, and unclear risk of bias (Figure 7). Most items analyzed were classified as unclear risk of bias (23 of 50 classifications), followed by low risk of bias (19 of 50) and high risk of bias (8 of 50). The main reason for the high risk of bias was incomplete outcome data (attrition bias). The overall risk of bias for experimental studies was not determined because SYRCLE’s RoB tool does not establish how to assess the overall risk of bias.

### 3.7. Quality of Evidence

The overall quality of evidence according to GRADE was rated as very low for all outcomes, both in RCTs and experimental studies. The main reasons for downgrading evidence when considering SBP, DBP, and HR in RCTs were risk of bias and imprecision. For experimental studies, evidence for SBP, DBP, MAP, HR, and NO outcomes was downgraded due to risk of bias, indirectness, and imprecision problems. GRADE rating can be found in the Appendix A.

#### Additional Information

Authors were contacted by email for additional information (such as mean and standard deviation of blood pressure values for sham groups) for performing meta-analyses [7,8,28] and to obtain a better description of PBM treatment parameters [17]. However, none of these outreach attempts were successful.

## 4. Discussion

Hypertension is a global burden and the leading risk factor for several life-threatening diseases [3,5,6]. One of the biggest challenges is achieving sustainable control of this disease within the normal BP range [6]. In this context, PBM is currently being investigated as a therapeutic strategy for the treatment of hypertension due to its mechanisms of action on biological systems, which include increasing NO levels [8]. This is the first systematic review to gather evidence about PBM treatment effects on hypertension. An extensive database search was carried out that combined all RCTs and experimental studies comparing PBM therapy to control groups. Overall, experimental studies noted that PBM treatments can reduce SBP [7,8,30,31], DBP [8,10], MAP [8,10], and HR [8,10], both acutely (single PBM treatment) [8,9] and chronically (multiple PBM treatments) [7,10], also increasing NO levels in hypertensive rats [8,30,31], all with very low-certainty evidence.

PBM on hypertension has been investigated in literature since 2008 when the first RCT investigated the benefits of laser acupuncture effects on hypertension. Until now, there is still a limited number of RCTs, all with low methodological quality [28]. Analyzing the reviewed studies, 4 weeks of PBM treatments appeared to be already beneficial for BP values [29]. Different frequencies of treatment were also noted to be effective, as both studies that used PBM for one [15] and three sessions per week [17,29] had positive results. However, a lack of follow-up does not allow us to evaluate the persistence of the demonstrated effects. Additionally, the results need to be analyzed cautiously considering the studies that only made intra-group comparisons and due to the overall high risk of bias. It is challenging to outline the key PBM parameters due to the insufficient parameter reporting in several studies. Only one study reported the energy used (16 J) [28], and one reported the power (mW) [26], but it was described confusingly as the ‘average peak power’. The only parameter that seemed to have unanimity between RCTs was the use of infrared wavelength, with one study not reporting the wavelength used [17]. This strongly limits replicability between studies and choice of parameters for future investigations. It is well-known that PBM acts by a dose–response window that can be different depending on the outcome [16]. Lack of reporting the applied dosage limits future applications and makes it more difficult to establish the dose–response window of PBM for hypertension.

It is worth emphasizing that all included RCTs used laser acupuncture. The use of low-power lasers to stimulate acupuncture points is an alternative developed to conventional acupuncture with needles [32]. A study that utilized PBM treatments without application on acupuncture points compared to laser acupuncture on temporomandibular disorders demonstrated similar results [32]. Thus, laser acupuncture can be considered a method of application of PBM treatment, as the therapeutic benefits of using lasers is evident. Regardless of the application site, the dosage applied needs to be in the optimal dose–response window to stimulate the desired effects [33]. Therefore, the principles of PBM treatments apply to laser acupuncture as well.

The experimental studies examining PBM treatments on hypertension have been more recent than RCTs (first study by Tomimura et al., 2014), supposedly inspired to examine the mechanism and optimize clinical protocols [10]. From experimental study results, it seems that energies of 3.6 J and 5.6 J per point are both efficient in achieving hypotensive effects. Still, it is important to study different dosages of PBM treatments in the same scenario for establishing the dose–response window, especially considering that some animals did not respond to the 5.6 J energy [7,8]. Considering RCTs, laser acupuncture seems to be an interesting method of application for achieving BP control, but the certainty of the evidence was very low [17,29]. Analyzing experimental study results, methods of application that target big blood vessels (tail vein) [7,10] or large body areas (abdominal wall) [8,30,31] seem to be effective. Translating these results to clinical practice/human investigations, methods such as intravascular laser irradiation of blood (ILIB), PBM treatment blankets, or whole-body PBM treatment devices could be relevant choices for future investigations.

The assessment of risk of bias of experimental studies was a challenge, as most studies did not report methodology details adequately. Most items assessed on the SYRCLE’s risk of bias tool were classified as unclear, due to a lack of clarity in reporting by the authors. Future experimental studies should use SYRCLE’s risk of bias tool as a guide for reporting their studies. For RCTs, it was also observed that many RoB 2 domains were classified as high risk of bias due to poor reporting of methodology. In this sense, as expected, the studies also did not report enough PBM treatment parameters, especially in RCTs. This limits direct comparisons between studies because different choices of parameters can result in completely different results. One important thing was that most RCTs used infrared wavelengths, while the red wavelength was the most investigated wavelength in experimental studies. Since in both scenarios, the therapies have shown interesting results, we suggest that future studies compare different wavelengths and also investigate the use of both wavelengths at the same time.

Overall, despite seeing some positive results, it is not possible to definitively conclude therapeutic benefits of PBM treatments on hypertension due to the lack of high-quality available evidence. All outcomes were graded as having very low-certainty evidence. Studies on this topic are limited, and it was not possible to compare them directly due to differences in control groups (sham groups/control groups/anti-hypertensive-only groups), making it impossible to perform robust meta-analyses and resulting in considerable heterogeneity (I^2^) in most of the meta-analysis. Ideally, we would like to compare studies that have similar control groups, which was not possible, given the small number of studies. For this reason, we suggest the results of the meta-analyses to be carefully interpreted due to the striking variation in the control groups. Also, this inability to group robust controls is the major reason the available evidence was graded as very low quality, preventing stronger conclusions. Due to different comparators, small number of studies, and lack of adequate reporting, it was only possible to perform a sensitivity analysis of the outcome SBP in experimental studies. In this analysis, exclusion of the only study that used an infrared wavelength decreased the heterogeneity to 0%. This can indicate that the use of different PBM parameters can also be responsible for the high heterogeneity obtained by our meta-analyses.

Adherence to the established therapies is a big challenge on hypertension. Up to 50% of patients do not adhere to antihypertensive medications. This poor adhesion can be related to social determinants, stress, or side effects related to medications [3]. When using PBM, strategies for improving adherence also need to be considered, although there is the advantage of not having any reported adverse effects.

Despite the lack of high-quality evidence, this is the first systematic review that gathered evidence on the role of PBM treatments on hypertension. This therapeutic strategy represents an innovative, low-cost, non-invasive, and non-pharmacological approach. This review can help researchers and clinicians appreciate the gaps in literature and direct future research on examining the efficacy of PBM treatments on hypertension. It is hoped that the results of this study will allow researchers and professionals to explore the possible adjunctive role of PBM so that in the future, better health services can be promoted for patients struggling with hypertension control. We highlight that there is still very low certainty on the available evidence, and more studies are needed to be able to draw conclusions about PBM effects on hypertensive patients. With the certainty of current evidence, it is not possible yet to clinically recommend PBM for hypertension. The major strength of this review is the methodology based on the Cochrane Handbook for Systematic Reviews of Interventions, along with the risk of bias assessment and the GRADE assessment. As putative future directions, we recommend pursuing high-quality RCTs with low risk of bias designs, including sham groups and follow-up periods, to investigate and standardize optimal PBM therapy dosing for hypertensive patients. Investigations on different wavelengths and different sites of application, such as ILIB therapy and whole-body PBM therapy, remain an open field for research.

## 5. Conclusions

PBM treatments have the potential to be an adjunct therapy for the treatment of hypertension, with studies showing a possible reduction in SBP, DBP, MAP, and HR, but the evidence is of very low certainty, coming from RCTs with high risk of bias and experimental studies with unclear risk of bias. The current evidence needs to be significantly improved with rigorous, well-designed experimental and clinical studies.

## Figures and Tables

**Figure 1 jcm-14-06716-f001:**
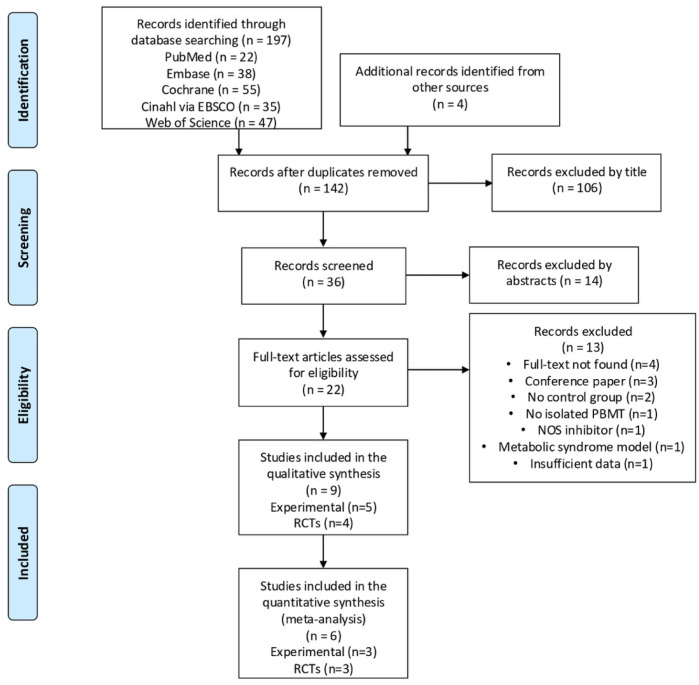
Prisma flow diagram.

**Figure 2 jcm-14-06716-f002:**
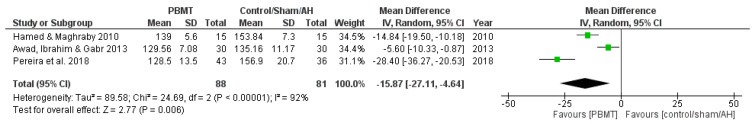
Comparison of photobiomodulation therapy versus control/sham/antihypertensive therapies on the outcome of systolic blood pressure for randomized controlled trials (sustained protocols), in mmHg. PBMT: photobiomodulation therapy; AH: antihypertensive. Favours [PBMT]: favors photobiomodulation therapy group. Favours [control/sham/AH]: favors control/sham/antihypertensive therapy group. SD: standard deviation. IV: inverse variance. CI: confidence interval [15,17,29].

**Figure 3 jcm-14-06716-f003:**
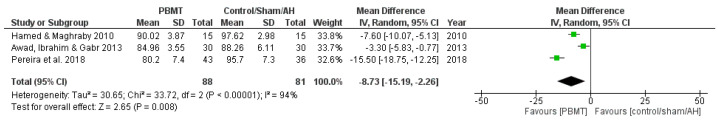
Comparison of photobiomodulation therapy versus control/sham/antihypertensive therapies on the outcome of diastolic blood pressure for randomized controlled trials (sustained protocols), in mmHg. PBMT: photobiomodulation therapy; AH: antihypertensive. Favours [PBMT]: favors photobiomodulation therapy group. Favours [control/sham/AH]: favors control/sham/antihypertensive therapy group. SD: standard deviation. IV: inverse variance. CI: confidence interval [15,17,29].

**Figure 4 jcm-14-06716-f004:**
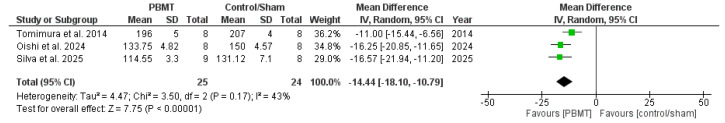
Comparison of photobiomodulation therapy versus control/sham on the outcome of systolic blood pressure for experimental studies (sustained protocols), in mmHg. PBMT: photobiomodulation therapy. Favours [PBMT]: favors hotobiomodulation therapy group. Favours [control/sham]: favors control/sham group. SD: standard deviation. IV: inverse variance. CI: confidence interval [10,30,31].

**Figure 5 jcm-14-06716-f005:**
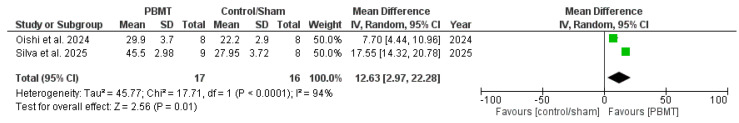
Comparison of photobiomodulation therapy versus control/sham on the outcome of serum nitrite and nitrate (NOx) for experimental studies (sustained protocols), in μM. PBMT: photobiomodulation therapy. Favours [PBMT]: favors hotobiomodulation therapy group. Favours [control/sham]: favors control/sham group. SD: standard deviation. IV: inverse variance. CI: confidence interval [30,31].

**Figure 6 jcm-14-06716-f006:**
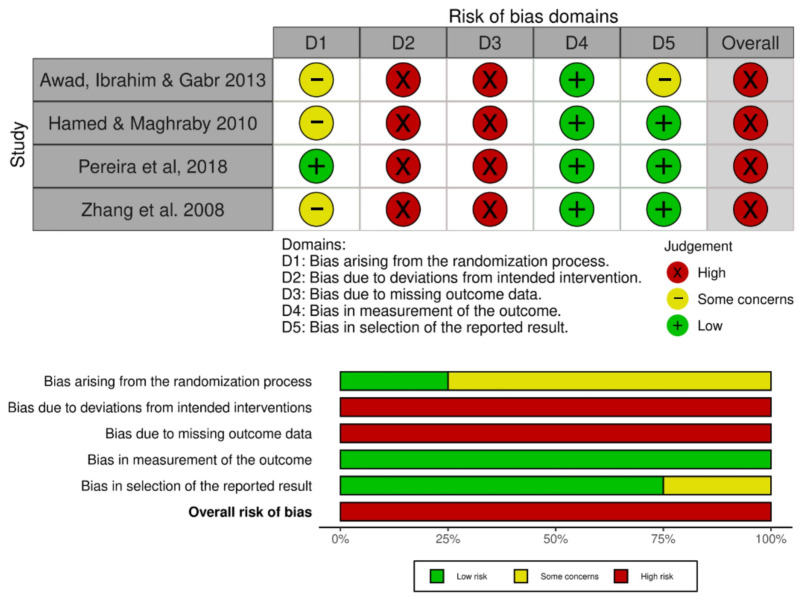
Risk of bias assessment of randomized controlled trials using the RoB 2 tool. The table on top shows the judgment for each study across the five domains: D1, bias arising from the randomization process; D2, bias due to deviations from intended interventions; D3, bias due to missing outcome data; D4, bias in measurement of the outcome; and D5, bias in selection of the reported result. Judgments are represented by round symbols: green (low risk), yellow (some concerns), and red (high risk). Overall risk of bias for each study is presented in the last column. The summary plot (bottom) illustrates the proportion of studies rated as low risk, some concerns, or high risk for each domain and for the overall risk of bias [15,17,28,29].

**Figure 7 jcm-14-06716-f007:**
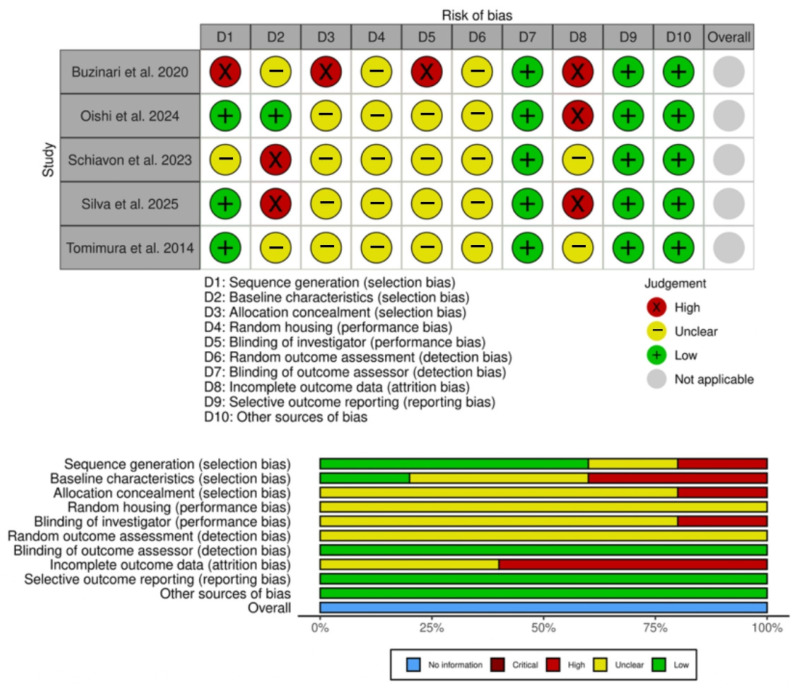
Assessment of risk of bias of experimental studies using the SYRCLE tool. The table on top represents the judgment for each study across ten domains: D1, sequence generation (selection bias); D2, baseline characteristics (selection bias); D3, allocation concealment (selection bias); D4, random housing (performance bias); D5, blinding of investigator (performance bias); D6, random outcome assessment (detection bias); D7, blinding of outcome assessor (detection bias); D8, incomplete outcome data (attrition bias); D9, selective outcome reporting (reporting bias); and D10, other sources of bias. Judgments are represented by round symbols: green (low risk), yellow (unclear), red (high risk), and gray (not applicable). The summary plot (bottom) shows the proportion of studies rated as low risk, unclear, high risk, or not applicable for each domain [7,8,10,30,31].

## Data Availability

All data generated by this review are available in the current manuscript or in the Appendix A.

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
