# Peer review of "Photobiomodulation Therapy in Hypertension Management—Evidence from a Systematic Review and Meta-Analysis"

_jcm, 2025, doi:10.3390/jcm14196716_

Round 1

Reviewer 1 Report

Comments and Suggestions for Authors

The topic is timely and relevant given the interest in non-pharmacological strategies for hypertension. However, the manuscript needs substantial improvements in clarity, consistency, and depth.

Introduction
Provides sufficient background and references, but some parts are overly long and repetitive (e.g., mechanisms of PBM). Clarify why PBM for hypertension specifically is worth a systematic review now.

Methods

Protocol registration on PROSPERO is a strength. However search strategy should be presented in a more concise way; currently too verbose.
Selection/exclusion criteria are clear, but justify why only RCTs and animal models were included (excluding quasi-experimental or pilot studies).
Detail how disagreements between reviewers were resolved beyond “by discussion.”
Risk of bias tools (RoB2, PEDro, SYRCLE) are appropriate, but their application needs more explanation.

Results
Flow diagram is appropriate. Tables are informative but too dense; consider summarizing or moving part of the detail to supplementary files.
Some forest plots are difficult to interpret (figures lack clarity in legends, abbreviations, and axis labels).
Reporting of meta-analysis lacks consistency: provide effect sizes, CIs, heterogeneity (I²), and certainty grading together.

Discussion

Strengths: situates findings in context, highlights novelty of PBM.

Weaknesses:

  • Conclusions are too strong given the “very low” certainty of evidence. Must be rephrased to avoid overstating clinical utility.
  • Discussion repeats large parts of results (SBP, DBP findings) instead of synthesizing.
  • Methodological limitations (small number of RCTs, heterogeneity of PBM protocols, high risk of bias, lack of follow-up) should be highlighted earlier and more prominently.
  • Consider shortening mechanistic speculations and focus more on clinical implications and research gaps.

Conclusions
Should be reframed as PBM shows potential but current evidence is scarce, low quality, and insufficient for clinical recommendations

Comments on the Quality of English Language

  • English is understandable but requires professional editing for grammar, conciseness, and consistency.

  • Many sections are repetitive and verbose (especially Methods and Discussion).

Author Response

Dear Reviewer 1,

We would like to sincerely thank you for the careful review of our manuscript. All suggestions and comments were thoroughly considered. We present here, very respectfully, a detailed point-by-point response to each one of your comments, along with the revised and improved manuscript.

Please see the responses.

Introduction

Provides sufficient background and references, but some parts are overly long and repetitive (e.g., mechanisms of PBM). Clarify why PBM for hypertension specifically is worth a systematic review now.

A: We thank the reviewer for the important comment. The introduction was reviewed and reduced, especially PBM mechanisms and the objectives of the review. The worthiness of the review was detailed and highlighted in the last paragraph of the introduction (page 2, lines 68-74).

Methods

Protocol registration on PROSPERO is a strength. However search strategy should be presented in a more concise way; currently too verbose.

A: Thank you for your suggestion. We have considered presenting the search strategy in a more concise way. However, the Preferred Reporting Items for Systematic reviews and Meta-Analyses (PRISMA) guidelines recommends that the full search strategy is described in systematic reviews. The presented search strategy was designed contemplating chapter 4 in the Cochrane Handbook for Systematic Reviews of Interventions, using a broad set of search terms related to the health condition of interest, the intervention and the study design to be included, trying not to miss any study that could be eligible.

References:

Page MJ, McKenzie JE, Bossuyt PM, et al. The PRISMA 2020 statement: an updated guideline for reporting systematic reviews. BMJ. 2021;372:n71. Published 2021 Mar 29. doi:10.1136/bmj.n71;

Higgins JPT, Thomas J, Chandler J, Cumpston M, Li T, Page MJ, Welch VA (editors). Cochrane Handbook for Systematic Reviews of Interventions version 6.5 (updated August 2024). Cochrane, 2024. Available from www.cochrane.org/handbook.

Selection/exclusion criteria are clear, but justify why only RCTs and animal models were included (excluding quasi-experimental or pilot studies).

A: We thank you for your question. When considering efficacy of interventions, the Cochrane Handbook for Systematic Reviews of Interventions points that a randomized trial is the preferred design for studying the effects of healthcare interventions, being the study design that is least likely to be biased. So, this choice was made thinking about reducing potential risk of bias derived from non-randomized studies. The presence of a control/placebo/usual care group is also important in the design to be sure that the effectiveness of the intervention being tested is not related to other confounding factors, such as placebo effect or the natural history of the disease.

Pilot studies that were randomized and had a control group were still considered. Animal models were also included, with a separate analysis, considering the lack of clinical trials in this field and the importance of animal studies to guide future clinical trials.

This information was added in page 3, lines 88-89.

Reference:

Higgins JPT, Thomas J, Chandler J, Cumpston M, Li T, Page MJ, Welch VA (editors). Cochrane Handbook for Systematic Reviews of Interventions version 6.5 (updated August 2024). Cochrane, 2024. Available from www.cochrane.org/handbook.

Detail how disagreements between reviewers were resolved beyond “by discussion.”

A: We thank you for noticing that this information was not clear. This information was added both in page 4 (lines 122-124) and page 5 (lines 167-169).

Risk of bias tools (RoB2, PEDro, SYRCLE) are appropriate, but their application needs more explanation.

A: We thank you for your suggestion. A more detailed explanation of how the tools were used was added to page 5 (lines 147-149, 157-159, 165-166).

Results

Flow diagram is appropriate. Tables are informative but too dense; consider summarizing or moving part of the detail to supplementary files.

A: We thank you for your suggestion and agree that the tables were too dense. Tables 2 and 3 were summarized, as can be found on Appendix A.

Some forest plots are difficult to interpret (figures lack clarity in legends, abbreviations, and axis labels).

A: We thank you for noticing that. All forest plots were reviewed, and the legends and abbreviations were rewritten in a more detailed manner.

Reporting of meta-analysis lacks consistency: provide effect sizes, CIs, heterogeneity (I²), and certainty grading together.

A: We thank you for your suggestion. The reporting for every meta-analysis was updated, and the certainty of the evidence was also added to each outcome of the systematic review.

Discussion

  • Conclusions are too strong given the “very low” certainty of evidence. Must be rephrased to avoid overstating clinical utility.

A: We thank you for noticing that. We have updated the conclusions and discussion removing/editing parts that overstated the results and adding new ones to emphasize the very low certainty of evidence. All modifications are highlighted in yellow.

  • Discussion repeats large parts of results (SBP, DBP findings) instead of synthesizing.

A: We thank you for noticing that. We have reduced the repetition of results in the discussion, and we highlight that in page 14 the results were repeated in a brief way to synthesize the main findings of the review (lines 480-483).

  • Methodological limitations (small number of RCTs, heterogeneity of PBM protocols, high risk of bias, lack of follow-up) should be highlighted earlier and more prominently.

A: We thank you for your suggestion. These limitations were added more prominently (page 14, lines 486-487). The very low certainty of evidence was also added in new paragraphs (lines 483, 520, 541, 562). We highlight that the lack of follow-up was emphasized in the first paragraph that the RCTs are mentioned (page 14, line 491), and the heterogeneity of PBM protocols was first mentioned in the same paragraph (page 14, lines 493-498).

  • Consider shortening mechanistic speculations and focus more on clinical implications and research gaps.

A: We thank you for your suggestion. The whole discussion section was reviewed to emphasize the clinical implications of the work and the current research gaps.

Conclusions

Should be reframed as PBM shows potential but current evidence is scarce, low quality, and insufficient for clinical recommendations

A: We thank you for noticing that. The conclusions section was updated emphasizing the very low certainty of evidence.

  • Many sections are repetitive and verbose (especially Methods and Discussion).

A: We thank you for noticing that. The Methods and Discussion sections were reviewed to improve the writing.

Reviewer 2 Report

Comments and Suggestions for Authors

This systematic review and meta-analysis on the effects of photobiomodulation (PBM) therapy in hypertension demonstrated that PBM therapy has the potential to be an adjunct therapy for the treatment of hypertension, being able to reduce systolic blood pressure, diastolic blood pressure, mean blood pressure and heart rate. Notably, the clinical trials noted PBM treatments reduced systolic blood pressure, diastolic blood pressure, and heart rate, but with very low certainty. Experimental lab studies corroborated PBM treatments reduced SBP, DBP, and mean arterial pressure while increasing nitric oxide levels, again with very low certainty.

The manuscript is well structured and focuses on the effects of PBM on hypertension. It is original, the statistical analysis is adequate, the tables and figures are clear, the references are appropriate, and the conclusions are well supported by the authors’ findings.

The manuscript is well suited for this Special Issue, which aims to highlight the advances of PBM in disease management.

Author Response

Dear Reviewer 2,

We would like to sincerely thank you for the careful review of our manuscript, and for acknowledging the novelty and importance of our review.

Kind regards.

Reviewer 3 Report

Comments and Suggestions for Authors

The topic of the article is timely and relevant, addressing a novel, non-pharmacological approach to hypertension. The systematic review is comprehensive and well structured. However, the certainty of evidence is very low, and this limitation should be more clearly and consistently emphasized throughout the paper. There are other issues that need further consideration:

1)The objectives are clear, but the framing occasionally implies stronger conclusions than the evidence allows. The authors should keep interpretations cautious.

2)Search and selection methods are sound, but some details (e.g., handling of missing data, criteria for “responsive animals”) could be clarified for transparency.

3)The unclear risk of bias across most included studies is a major limitation. This should be more prominently highlighted in the abstract and conclusions. Also,statements on efficacy should be tempered by repeating the “very low certainty” grading.

4)The authors should consider discussing more explicitly how heterogeneity in PBM parameters (dose, wavelength, application sites) limits comparability and synthesis.

5)While the discussion raises possible applications (e.g., intravascular PBM, whole-body devices), these remain speculative. The section would benefit from a more balanced tone. This section could be shortened to focus more on PBM in hypertension rather than broader PBM mechanisms.

6)Regarding the figures and tables the authors should complement clearer labeling of key findings would help readers.

7)The authors should include the most recent hypertension guidelines to contextualize clinical relevance.

8)The authors should revise the manuscript for mistypo.

Author Response

Dear Reviewer 3,

We would like to sincerely thank you for the careful review of our manuscript. All suggestions and comments were thoroughly considered. We present here, very respectfully, a detailed point-by-point response to each one of your comments, along with the revised and improved manuscript.

Please see the attachment for the responses.

However, the certainty of evidence is very low, and this limitation should be more clearly and consistently emphasized throughout the paper.

A: We agree and thank you for your suggestion. We have added this information on each outcome’s results, in new parts of the discussion and in the conclusions. All modifications are highlighted in yellow in the manuscript.

1)The objectives are clear, but the framing occasionally implies stronger conclusions than the evidence allows. The authors should keep interpretations cautious.

A: We thank you for noticing that. The whole manuscript was reviewed to not overstate the current evidence.

2)Search and selection methods are sound, but some details (e.g., handling of missing data, criteria for “responsive animals”) could be clarified for transparency.

A: We thank you for your suggestion. Handling of missing data was added in page 4 (lines 135-138). We also clarify that the “responsive animals” was a stablished criteria by authors of the included studies, not by the systematic review authors. This information was added in page 9 (lines 299 and 324) and 10 (lines 336 and 361).

3)The unclear risk of bias across most included studies is a major limitation. This should be more prominently highlighted in the abstract and conclusions. Also, statements on efficacy should be tempered by repeating the “very low certainty” grading.

A: We thank you for your suggestions. The risk of bias result was added to the abstract (lines 24-29) and conclusions (page 16, lines 575-576). The very low certainty grading was also added on each outcome’s results, in new parts of the discussion and in the conclusions.

4)The authors should consider discussing more explicitly how heterogeneity in PBM parameters (dose, wavelength, application sites) limits comparability and synthesis.

A: We thank you for your suggestion and we strongly agree that this should be more discussed. This discussion was added in pages 14 (lines 498-502) and 15 (lines 532-534).

5)While the discussion raises possible applications (e.g., intravascular PBM, whole-body devices), these remain speculative. The section would benefit from a more balanced tone. This section could be shortened to focus more on PBM in hypertension rather than broader PBM mechanisms.

A: We thank you for your suggestion. The paragraph was rewritten to add a more balanced tone (page 15, lines 524-525).

6)Regarding the figures and tables the authors should complement clearer labeling of key findings would help readers.

A: We thank you for your suggestion. All figures and tables were built and described to report in detail all results and then sustain all discussion, which highlighted all the key findings. 

7)The authors should include the most recent hypertension guidelines to contextualize clinical relevance.

A: We thank you for your important suggestion. The most recent hypertension guidelines from the European Society of Hypertension, American College of Cardiology/American Heart Association were added to the contextualization (page 1 – lines 37-39, page 2 – lines 46-47).

8)The authors should revise the manuscript for mistypo.

A: We apologize for any mistypos that could have been missed in the text. We have reviewed the whole manuscript and checked carefully to reduce typing errors.

Reviewer 4 Report

Comments and Suggestions for Authors

As a hypertension specialist, I find the topic of this study very exciting because it is completely unknown to me and, I am sure, to many of my colleagues in the field.

This is the first and, in my opinion, greatest strength of the study, which does not meet my criteria for a meta-analysis with 3/3 included studies.  

Overall, it is striking that the affiliation of the authors overlap with those of the authors of the analyzed studies. Given the small number of studies included, this leaves a certain impression. A meta-analysis of three studies from a single sphere of influence is scientifically questionable. 

It is imperative to explain why, for example, the experimental study by Oishi et al. Life Sciences 2017 is mentioned in the discussion but not included in the analysis.

Due to the small number of original studies available, the significance remains extremely limited. The authors make this clear in the discussion, so that readers and I as a reviewer can live with it. Overall, this discussion is very well done, especially since it also discusses the aspect that the treatment was performed as laser acupuncture.  

I do see the following major points: 

Introduction:      

It must be explained why the effectiveness of antihypertensive therapy is low: compliance is the major problem here - this must be identified and explained. Laser acupuncture will not help in this case, as it is likely that only interested and therefore compliant patients would use it. This must be discussed. With adequate medication and patient adherence, any hypertension can be adequately treated in everyday clinical practice.

Results: 

The actual results of the individual studies are presented in a very vague manner in the results section; this narrative style is not helpful. Since there are only three studies, the extent to which blood pressure was lowered can be presented in numerical values. 

"Side effects None of the included studies reported side effects associated with PBM treatments. (Line 337/338)"
What side effects were investigated in the animals? What parameters were used to evaluate them?  

“The authors were contacted by email (...). However, none of these attempts to contact them were successful.”
This is hard to believe, since the authors of the studies cited (at least according to the affiliations listed in the papers) work at the same university as the authors of this review. 

Discussion

“In this context, PBM treatment is a promising strategy for the treatment of hypertension due to its therapeutic effects, especially given the increase in NO levels achieved by this therapy.”
- The studies presented by the authors strongly challenge this clear assumption—if this were the case, the studies would have been published in other journals and the therapy would be part of routine clinical practice.
- This needs to be reworded more carefully.

Minor point:

Adverse effects in line 266 should be in italics. 

Author Response

Dear Reviewer 4,

We would like to sincerely thank you for the careful review of our manuscript. All suggestions and comments were thoroughly considered. We present here, very respectfully, a detailed point-by-point response to each one of your comments, along with the revised and improved manuscript.

Please see the responses.

Overall, it is striking that the affiliation of the authors overlap with those of the authors of the analyzed studies. Given the small number of studies included, this leaves a certain impression. A meta-analysis of three studies from a single sphere of influence is scientifically questionable.

A: We thank you for noticing that and we agree that this may leave a certain impression. However, the research group mentioned (authors of 4 of the 9 included studies) have no direct contact with our research group and had no influence on this systematic review. Federal University of São Carlos (UFSCar) is a big university in Brazil, with over 22 laboratories only in the Physical Therapy Department (our department) and over 8 laboratories in the Physiological Sciences Department (department of the mentioned research group). UFSCar also has a great history/tradition on photobiomodulation therapy research, so the probability of reviews about this therapy to include studies from UFSCar is high, justifying the mentioned sphere. We also clarify that we have done our best effort to build and describe a comprehensive search strategy, clear inclusion criteria and detailed supplementary materials, trying to make it clear that any published study that met the established criteria would be included.

It is imperative to explain why, for example, the experimental study by Oishi et al. Life Sciences 2017 is mentioned in the discussion but not included in the analysis.

A: We thank you for noticing that. We had wrongly cited Oishi et al. 2017 in the discussion section, when meaning to cite Oishi et al. 2024 (included article). The error was corrected, and we have double checked the references and citations to make sure there are no more mistakes such as this one. We also clarify that the study by Oishi et al. 2017 was not included due to not having a control group, and this is described at the Supplementary File 1 (Table of excluded articles).

Due to the small number of original studies available, the significance remains extremely limited. The authors make this clear in the discussion, so that readers and I as a reviewer can live with it. Overall, this discussion is very well done, especially since it also discusses the aspect that the treatment was performed as laser acupuncture. 

A: We agree that the clinical significance is limited at this point, and we thank the reviewer for recognizing that the discussion made this clear.

Introduction:     

It must be explained why the effectiveness of antihypertensive therapy is low: compliance is the major problem here - this must be identified and explained. Laser acupuncture will not help in this case, as it is likely that only interested and therefore compliant patients would use it. This must be discussed. With adequate medication and patient adherence, any hypertension can be adequately treated in everyday clinical practice.

 A: We thank you for suggestion. This information was added in the introduction (page 2, lines 46-47) and discussed on page 15 (lines 550-554).

Results:

The actual results of the individual studies are presented in a very vague manner in the results section; this narrative style is not helpful. Since there are only three studies, the extent to which blood pressure was lowered can be presented in numerical values.

A: Thank you for your suggestion. We would like to clarify that for the outcomes Systolic Blood Pressure and Diastolic Blood Pressure for RCTs, and Systolic Blood Pressure and Nitric Oxide for experimental studies, meta-analyses were performed. All meta-analyses include the mean difference considering all included studies for the outcome. The forest plot also includes the mean values and mean differences for each individual study. For outcomes that we could not perform meta-analyses, the individual values for included studies were now added (page 8, lines 281-283; page 9, lines 320-321, 324-326 and 332-333; page 10, lines 336-338, 344-345 and 346-347).

"Side effects None of the included studies reported side effects associated with PBM treatments. (Line 337/338)"

What side effects were investigated in the animals? What parameters were used to evaluate them? 

A: Thank you for your question. No studies reported specific methods to investigate adverse effects, this information was added in page 8, line 287, and page 11, line 378.

“The authors were contacted by email (...). However, none of these attempts to contact them were successful.”

This is hard to believe, since the authors of the studies cited (at least according to the affiliations listed in the papers) work at the same university as the authors of this review. 

 A: Authors from the included studies Schiavon et al. 2023; Buzinari et al. 2020.; Zhang et al. 2008.; and Awad, Ibrahim & Gabr, 2013 were all contacted asking for missing data from the published articles. Authors of two of the articles are affiliated to Federal University of São Carlos (UFSCar). Still, UFSCar is a big university, and our laboratory has no direct contact with the author’s laboratories. Correspondence authors were contacted by email, but the asked data was not sent, as shown by the attached picture below (automatic translation of the webpage by Google was used, since the emails were sent in Portuguese).

Round 2

Reviewer 1 Report

Comments and Suggestions for Authors

Dear authors,
Thank you for improving and correcting the manuscript and for your clarifications. However, several points need to be addressed: 

Introducction

Properly contextualise PBM and hypertension; justify the need for a formal synthesis; cite the PRISMA/PROSPERO framework.
The following points could be addressed:

- Clarify the gap: explain in the last paragraph why a new review is necessary (e.g., recent trials, heterogeneous PBM parameters, combination of clinical and animal studies with the GRADE framework).

- Separate mechanisms vs. clinical rationale in 1–2 sentences to avoid repetition and maintain focus on the clinical question (effect on SBP/DBP).

Methods

PROSPERO registry, comprehensive search, defined PICO criteria; appropriate bias tools (RoB2/SYRCLE); quantitative synthesis separated by domains.
Would be good to:

- Comment on the search strategy: move excessive details to the Supplement and leave only equations/dates/keywords in the main text (for readability).

- Meta-analysis: in the statistics section, indicate a priori the model (random), the metrics (MD/SMMD), how heterogeneity was handled (I²) and whether there was meta-regression or subgroup analysis (e.g., wavelength, sham vs. standard comparator).

- Publication bias: specify whether funnel/Egger tests were performed and the limitations due to small n; if not applicable, state this explicitly.

- GRADE: detail the domains that led to very low certainty (inconsistency, imprecision, RoB) and link them to each outcome.

Results

Flow chart; separate synthesis of RCTs/animals; reduction in SBP/DBP with high recognised heterogeneity.
The following could be done:

- Make effect sizes, 95% CI and I² explicit in the text (not just in figures) for each meta-analysis; cite n of studies per comparison.

- PBM parameters: unify the table with wavelength, power, energy, exposure time, spot, dose (J/cm²) and application point; indicate ‘not reported’ when missing.

- Sensitivities: add analysis restricted to sham comparators and (if there is sufficient n) by wavelength or application pattern, to explore the origin of I².

Discussion

Cautious tone; acknowledges limitations (high RoB, heterogeneity, very low certainty). The following should be added:

- Explore the sources of heterogeneity in greater depth: discuss the differences in dosimetry/target points and their physiological plausibility; link to the parameter table.

- Emphasise that, with this certainty, PBM does not yet allow for clinical recommendations.

- Discuss future lines of research (prioritise trials with sham, standardised dosimetry and follow-up ≥12 weeks, and additional outcomes).

Conclusions

Moderate conclusion consistent with GRADE. Congratulations.

Author Response

Answer to Reviewer 1 - report 2

Dear Reviewer 1,

We sincerely thank you for the time and effort dedicated to reviewing our manuscript. We have carefully considered all your suggestions. We present here, very respectfully, a detailed point-by-point response to each one of your comments, along with the revised version of the manuscript.

Introduction

The following points could be addressed:

- Clarify the gap: explain in the last paragraph why a new review is necessary (e.g., recent trials, heterogeneous PBM parameters, combination of clinical and animal studies with the GRADE framework).

A: We thank the reviewer for the important suggestion. A clearer explanation of the gap in the literature was added in the introduction (page 2, lines 73-76).

- Separate mechanisms vs. clinical rationale in 1–2 sentences to avoid repetition and maintain focus on the clinical question (effect on SBP/DBP).

A: We thank the reviewer for the suggestion. The mechanisms vs clinical rationale was now emphasized in page 2 (lines 57-61).

Methods

Would be good to:

- Comment on the search strategy: move excessive details to the Supplement and leave only equations/dates/keywords in the main text (for readability).

A: We thank you for the suggestion and agree that this can improve readability. The table with the search terms was moved to the Supplementary File.

- Meta-analysis: in the statistics section, indicate a priori the model (random), the metrics (MD/SMMD), how heterogeneity was handled (I²) and whether there was meta-regression or subgroup analysis (e.g., wavelength, sham vs. standard comparator).

A: We thank you for the suggestion. We have added a more detailed explanation about the statistics methods (page 5, lines 180-188). About subgroup analyses, the collected data from studies was carefully reviewed, but unfortunately these analyses were not possible due to the small number of studies included in each meta-analysis (maximum of three studies), along with the great heterogeneity between them.

Example: For RCT’s meta-analyses, two of the studies used infrared wavelengths, while the other one did not report the used wavelength. For the same comparisons, one of the studies used simulated photobiomodulation (PBM) along with medication use (Pereira et al. 2018), while other study used just medication without simulated PBM (Awad, Ibrahim & Gabr, 2013), and the other one only had a control group without medication or simulated PBM (Hamed & Maghraby, 2010).

The sensitivity analysis was added for the outcome SBP in experimental studies, which was the only outcome that included a sufficient number of studies with distinct wavelengths, allowing us to analyze if the removal of the only study that used an infrared wavelength would change the heterogeneity. This information was added in page 5, lines 186-188.

- Publication bias: specify whether funnel/Egger tests were performed and the limitations due to small n; if not applicable, state this explicitly.

A: We thank you for the suggestion. Unfortunately, it was not possible to perform Funnel Plots/Egger tests due to the small number of included studies. It is indicated by the Cochrane Handbook for Systematic Reviews that there should be at least 10 studies included in the meta-analysis to perform funnel plots analysis. This non applicability was added in page 5 (lines 184-186).

- GRADE: detail the domains that led to very low certainty (inconsistency, imprecision, RoB) and link them to each outcome.

A: We thank you for the suggestion. We have added a detailed description of GRADE domains in the methods section (page 5, lines 197-203) and a more detailed explanation about GRADE results, relating them to the outcomes (page 13, lines 483-486).

Results

The following could be done:

- Make effect sizes, 95% CI and I² explicit in the text (not just in figures) for each meta-analysis; cite n of studies per comparison.

A: We thank you for the suggestion. The meta-analysis results descriptions were reformulated, and the n of studies was also added (page 7, lines 253-257 and 277-280; page 8, line 281; page 9, lines 321-327; page 10, lines 384-387).

Additionally, we would like to inform you that the year of publication of the study by Pereira et al. was entered incorrectly in Figures 2 and 3 and has now been corrected.

- PBM parameters: unify the table with wavelength, power, energy, exposure time, spot, dose (J/cm²) and application point; indicate ‘not reported’ when missing.

A: We thank you for the suggestion. The tables for PBM characteristics were unified (Appendix B) and the “not reported” indication was also added.

- Sensitivities: add analysis restricted to sham comparators and (if there is sufficient n) by wavelength or application pattern, to explore the origin of I².

A: We thank you for the suggestion. We truly believe that a sensitivity analysis for every outcome would improve the quality of our review. However, due to the small number of studies and differences in the comparator groups, it was only possible to perform a sensitivity analysis for the outcome SBP in the experimental studies.

 As we have mentioned, for RCTs, the three included studies used different comparator groups, while one of the authors did not report the PBM wavelength that was applied. As for experimental studies, two of the included studies (Tomimura et al. 2014 and Silva et al. 2015) had sham groups, while Oishi et al. 2024 does not make it clear if their control group received sham therapy or not. Since only one of the experimental studies (Tomimura et al. 2014) used an infrared wavelength, the sensitivity analysis was performed removing this group from the meta-analysis. This information was added in the methods section (page 5, lines 186-188) and in the results section (page 9, lines 324-327).

Discussion

The following should be added:

- Explore the sources of heterogeneity in greater depth: discuss the differences in dosimetry/target points and their physiological plausibility; link to the parameter table.

A: We thank you for the suggestion. The sources of heterogeneity were now added in more detail on page 15 (lines 565-568; 573-578). We also want to highlight that sources of heterogeneity and differences related to PBM parameters were also previously mentioned throughout the discussion (page 14, lines 516-525; page 15, lines 539-548 and 554-561).

- Emphasise that, with this certainty, PBM does not yet allow for clinical recommendations.

A: We thank you for the suggestion. This was emphasized in page 16 (lines 593-594).

- Discuss future lines of research (prioritise trials with sham, standardised dosimetry and follow-up ≥12 weeks, and additional outcomes).

A: We thank you for the suggestion. The future directions were added in more detail in page 16, lines 597-601.

Reviewer 4 Report

Comments and Suggestions for Authors

We thank the authors for attempting to address the points raised above. However, the included studies exhibit such serious scientific shortcomings that a meta-analysis based on them cannot yield reliable conclusions. Furthermore, the process of selecting which studies to include remains unclear to me and is therefore not reproducible. An analysis of arbitrarily selected studies is simply not equivalent to a systematic meta-analysis.   

Author Response

Answer to Reviewer 4 - report2

Dear Reviewer 4,

We would like to thank you for all your efforts and time dedicated to review our manuscript. We tried one more time, and very respectfully, to bring a detailed point-by-point response to each one of your comments, along with the revised and improved manuscript.

However, the included studies exhibit such serious scientific shortcomings that a meta-analysis based on them cannot yield reliable conclusions.

A: Thank you for this important comment. Unfortunately, the evidence on the topic of the review does has serious limitations. On other hand, we know that the role of a systematic review is exactly to point out the status of the evidence, with your strengths and/ or limitations, for future decision make, or point out to the need of future studies with better methodological quality to improve the level of the evidence. And our manuscript fulfils this role based on the Cochrane Handbook for Systematic Reviews (the gold standard guideline for systematic reviews), chapter 7, section 7.6, which states that including only studies at low risk of bias may produce imprecise results. In addition, other strength of the present manuscript was that limitations from studies were analysed and pointed by appropriate tools (RoB 2, PEDro and SYRCLE), as the whole body of evidence was analysed by GRADE system. Finally, we recommended in our manuscript the need of more studies with better methodological quality in the future.

Furthermore, the process of selecting which studies to include remains unclear to me and is therefore not reproducible. An analysis of arbitrarily selected studies is simply not equivalent to a systematic meta-analysis.  

A: Thank you for this important comment. We tried to write all methods as clear as possible, respecting all recommendations from PRISMA and the Cochrane Handbook for Systematic Reviews, since both are considered gold standards for systematic reviews. Moreover, we have improved the methodology section with the specific suggestions from all reviewers in round 1 and 2 of the review process, and remain open to improving it, as we do value the highest quality for our manuscript. Respectfully, please, point out to us all the unclear aspects of the methodology described in the manuscript, and which does not match with the recommendations stated by PRISMA and the Cochrane Handbook for Systematic Reviews for corrections and improvements.